# New Parkrunners Are Slower and the Attendance Gender Gap Narrowing Making Parkrun More Inclusive

**DOI:** 10.3390/ijerph20043602

**Published:** 2023-02-17

**Authors:** Andre S. Gilburn

**Affiliations:** Biological and Environmental Sciences, University of Stirling, Stirling FK9 4LA, UK; andre.gilburn@stir.ac.uk

**Keywords:** mass participant event, obesity epidemic, barriers to exercise

## Abstract

Parkrun is a weekly mass-participation event. Finishes are recorded, with the resulting database potentially containing important public health information. The aim of this study was to identify characteristics of events that overcome barriers to participation, and to identify changing patterns in the demographics of participants. GLMMs were generated of age-graded performance, gender ratio and age of participants at Scottish parkrun events. Predictor variables included age, gender, participant, runs, date, elevation gain, surface and travelling time to the next nearest venue. There was a decline in the mean performance of participants at events, yet individual performances improved. The gender ratio showed higher male participation with a narrowing gender gap. Events in the most remote parts of Scotland had lower performance and a higher proportion of female participants. Events on slower surfaces had more female participants. Parkrun events are becoming more inclusive, with more females and participants exhibiting low performance. In more remote parts of Scotland, more females participated in parkrun than males, suggesting parkrun has overcome traditional barriers to female participation in sport. Prioritising the creation of events at remote locations and on slower surfaces could increase inclusivity further. General practitioners prescribing parkrun might want to prescribe attendance at slower events for female patients.

## 1. Introduction

The UK has been identified as one of the worst countries in the world for the activity level of its citizens by an international survey of inactivity, which found 30–40% of men and 40–50% of women in the UK undertake insufficient physical activity [1]. The gender gap in inactivity in the UK is mirrored by studies in Ireland [2,3]. Inactivity is a particular problem in Scotland, which has the highest level of morbid obesity in the UK [4]. Scotland has seen recent increases in the death rates of under 65s [5]. The level of morbid obesity is predicted to increase dramatically in the UK over the next two decades, so this is a worsening problem that requires urgent solutions [4].

One partial solution could be mass-participation sports events. Whether such events increase sporting activity is a matter of debate [3]. Annual long-distance events, such as half and full marathons, might change runners’ habits over the short term while training for the event, but participants might relapse once the event has finished [3]. A different model of mass-participation event is operated by parkrun, who provide free weekly 5-kilometre (km) events at fixed venues across more than 20 countries [6]. There is no requirement to run at parkrun, with wheelchairs, mobility scooters, buggies and dogs welcomed at the majority of 5 km events. Parkrun also actively encourages walking, and many events have a volunteer parkwalker role [6]. This aims to make parkrun more inclusive than more traditional races. Indeed, it is stressed that parkrun is not a race, so it is potentially less intimidating to the inactive. A year-long study of new participants at a UK parkrun event showed an increase in fitness and an associated reduction in body weight [7]. This study also reported increases in perceived happiness and decreases in perceived stress [7]. Many people who did not consider themselves runners have subsequently become runners through participation at parkrun events [8]. These people also felt they benefitted the most from parkrun in a survey [8]. Another survey of participants with mental health issues found they felt that parkrun significantly improved their condition [9]. These studies show parkrun participants report multifaceted health benefits to participation.

To maximise its impact, parkrun UK has launched various initiatives to increase inclusivity and reach the sections of the community that would benefit the most from participating at events. For example, the parkrun practice links parkrun to medical practices in the UK, so that patients who would benefit from increasing their activity levels are being prescribed parkrun by general practitioners [10]. Parkrun have also targeted improving inclusivity via initiatives such as parkwalk and PROVE (the parkrun Running or Volunteer for Everyone initiative) which have helped persuade those with long-term health conditions to participate in parkrun. One indicator that parkrun is becoming increasingly inclusive is the finding that the proportion of participants with disabilities rose from 4.3% in 2013 to 9.2% in 2019 [11].

There are now more than 700 parkrun event venues in the UK and over 2000 worldwide, with Scotland hosting 56 weekly parkrun events by the end of 2019. Deprived areas of the UK have been found to have easier access to parkrun events [12]. Despite this, participation is higher in white, middle-class members of the community, with lower participation levels being seen in areas with greater density of ethnic minorities and more deprived areas, despite their greater access [13].

The number of males and females registering for parkrun is similar, but fewer females participate [14]. This suggests there are specific barriers to women participating in parkrun. As women are less active, identifying and removing these barriers would help maximise the potential beneficial impact of parkrun events. A study in Tasmania showed that parkrun attracted a broad range of participants, including those not currently participating in sufficient physical activity, with social factors being the most important driver of participation [15]. The weekly nature of parkrun means it has a greater contribution to weekly activity recommended targets, with the Tasmanian study finding those taking up parkrun included many non-runners, the obese and those who participated with their children. As parkrun allows children from age 4 to participate, and allows adults to push younger children at events in buggies, including double buggies, it potentially removes one barrier to participation by enabling parents to participate with their children [15]. As women are likely to have greater childcare responsibilities, this might be particularly important for attracting female participants [15].

Scotland has some of the most deprived areas of the UK, with some of the greatest health problems and levels of morbid obesity [4,5]. Scotland also hosts some of the UK’s most geographically isolated communities, where social factors might result in a different profile of parkrun participants. Several isolated communities in Scotland now host parkrun events, such as Kirkwall in Orkney, Bressay in Shetland, Thurso in Caithness and Mount Stuart on Bute. The variety of locations of parkrun events in Scotland makes it an excellent potential model system to demonstrate how well parkrun is impacting fitness levels in different types of communities. Previous studies of parkrun participants and gains in fitness and the benefits of parkrun have predominantly been based upon surveys and following small cohorts of participants [16,17].

All finishes at parkrun events are published on their website [18]. Results from participation in parkrun events are provided as a time and an age-graded performance score, (AGPS) which is adjusted for both the age and gender of a participant to allow direct comparisons between all [17]. One study used AGPS to compare first parkrun performance against maximum performance for different cohorts [17], revealing that non-runners improved their performance the most from attending parkrun. No previous studies have used general linear mixed modelling (GLMMs) to analyse patterns in AGPS. GLMMs allow analysis of large noisy datasets with large numbers of covariates, and where there are repeated measures and non-independence within a grouping variable and where there are large differences in sample sizes between levels of that grouping variable [19,20]. They allow the separation of within- and between-group variance. For the parkrun results dataset, crucially, GLMMs will allow the separation of within and between participant variation in performance, allowing a separate measure of changes in the overall performance of the demographic of parkrunners, and also of the changes in performance of individual parkrunners over time [19,20]. The aim of the current study was to conduct the first quantitative assessment of AGPS data from all the parkruns in one area to determine if the population of parkrunners is changing in performance level over time, whether new participants are changing in their initial performance levels over time and whether there are changes in the gender ratio and age of participants over time. Identifying changing patterns in performance both within and between participants could provide important information about the public health benefits of parkrun. Scotland was chosen as the study area, as it exhibits a great variety of parkrun locations and has established problems with obesity, making it an ideal model system for studying the impact of parkrun on public health. In addition, an assessment of participation levels was conducted to determine whether location-based factors affected engagement with parkrun in Scotland and whether this varied with the characteristics of participants.

## 2. Materials and Methods

The study was an analysis of existing datasets. No new data were collected, nor were any participants involved in the study. Data were harvested from the parkrun results pages for all parkrun events that took place in Scotland from 6th December 2008, when the first event took place at Pollok Park in Glasgow, until the 1st January 2020 [18]. The mean number of participants at the 56 Scottish event venues ranged from 418.7 for Edinburgh to 26.2 at Girvan prom, and a total of 10,012 separate parkruns were included, ranging from 567 events held at Pollok to just 3 events at Agnew in Stranraer.

### 2.1. Participant Characteristics

The results page for each event was cut and pasted into an Excel file and processed using an Excel macro [21]. This extracted information about each participant included their age category, parkrun registration number, gender, AGPS, number of participations and the date. Details about these participant characteristics are provided in Table 1. Parkrun registration numbers are allocated to participants on a worldwide basis when they first register. There are now more than 7 million registered parkrunners. The numbers are allocated chronologically, so a higher number refers to a more recent registrant.

### 2.2. Event Characteristics

Three additional variables characterising various aspects of the different event venues that could affect performance and attendance were also included in the models (Table 1). These were elevation gained in metres completing each course, the surface type and the travelling time to the next nearest parkrun in minutes.

Elevation gain was determined by plotting the main parkrun route used at each event venue on Strava routes, which provides the number of metres both climbed and descended on a route based upon an elevation database [22]. This is different to the elevation measures collected on runs when using Strava, which can be inaccurate. At most event venues, the start and finish were close together, so there was no difference in metres climbed and descended. At event venues where the start and the finish were separated, there was not a substantial difference in elevation, with the maximum being 6 m. The mean elevation gained was 43 m with a range of 0 to 114 m. Using metres descended made no substantive difference to the outcomes of the models, and the two variables were too highly correlated to be included in the same model, so only elevation gain was used.

Parkrun events are run on a range of surface types. Hard surfaces, such as tarmac and concrete, have a relatively low ground contact time and are therefore faster to run on than softer surfaces, such as grass and trail paths. The information on the surface at each event venue was taken from the parkrun event course descriptions [18]. Courses run entirely on hard surfaces, such as concrete and tarmac, were assigned a surface score of 2, those on a mix of hard and softer surfaces a score of 1 and those on entirely softer surfaces a score of 0.

Travelling time between parkrun locations was determined in minutes using Google maps between the start location of each pair of event venues. The data were collected between 8 am and 9 am on a Saturday morning to obtain traffic conditions comparable to a parkrun day. Events in Scotland start at 9.30 am, half an hour later than in the rest of the UK. The shortest time in minutes to another event venue was used as a measure of how isolated an event venue was.

### 2.3. Age Graded Performance Scores

AGPS was used as the measure of how well a parkrunner performed at an event [17]. This is a measure of athletic performance relative to the world record for that age group and gender [23]. As it is adjusted to take into account both age and gender, it enables direct comparisons across all participants, and is presented as a % of the world record performance [24].

### 2.4. Data Analysis

The data consisted of the results of 1,572,104 participations by 167,440 parkrunners, consisting of 695,127 females and 915,668 males. The dataset was analysed in R x64 version 4.0.3. All numerical predictor variables were scaled to have a mean of zero and a standard deviation of one. Details of all predictor variables are provided in Table 1.

General linear mixed models (package lme4) were used to analyse AGPS and age [19,20]. A generalised binomial mixed model (package MuMIn) was used to analyse the gender of participants at events. Participant was used as a random effect, as it contained 167,440 levels, many of which only contained a single data point, and there was considerable variation in the number of samples within levels, with some participants taking part several hundred times. This means that participant could not be used as a fixed effect due to lack of degrees of freedom within some levels. As the effect sizes for random effects are generated using partial pooling participant can be used as a random effect. Indeed, mixed models and random effects were developed to handle grouping variables of exactly this type, where there are large numbers of levels and significant variation in the sample size of each level. The use of participant as a random effect in mixed models has been discussed at length in the literature, and is now common practice in some other fields, such as psychology [20].

Significance levels were determined using the package lmertest. Registration number was also used as a continuous variable, as numbers are allocated in chronological order to investigate variation in the performance of parkrunners with respect to the order that they registered for parkrun.

All figures were generated using ggplot2. Parkrun ID cohorts were created to enable visualisation of the results (although they were not used in the analyses) with participants split into groups based upon their number, with the first group having numbers 1–500,000, the second 500,001–1,000,000, etc., up to the final cohort of 6,500,001–7,000,000. These cohorts were not used in the models.

## 3. Results

### 3.1. Is Mean Performance at Parkrun Events in Scotland Declining?

A linear regression model showed a highly significant decline in performance over time (F = 17,960, d.f. = 1, 1,572,102, *p* < 0.001, parameter estimate = −1.011, s.e. = 0.008). The mean AGPS of participants at Scottish parkrun events has declined year on year from 2008 to 2019 (Figure 1).

### 3.2. Are New Registrants Getting Slower?

A linear regression model showed a highly significant negative association between performance and registration number (F = 137,815, d.f. = 1, 1,572,102, *p* < 0.001, parameter estimate = −2.937, s.e. = 0.008). When runners are separated into cohorts based upon registration number the continuing decline in performance of new registrants is clear (Figure 2).

### 3.3. What Factors Determine AGPS at Scottish Parkrun Events?

A linear mixed model shows that registration number is negatively correlated with AGPS (Table 2). Elevation gain was also strongly negatively correlated with performance. Males performed significantly better than females. Surface score was positively correlated with AGPS, showing that harder surfaces were associated with better performance. The number of parkruns completed and date of event were both positively correlated with performance, showing that individual performances improved over time and with number of participations. Age was also significantly associated with performance, with older runners performing better. The final significant predictor was the travelling time to the next nearest parkrun, with lower performance scores at the most remote parkrun venues. There were also three significant interaction terms maintained in the model. Two of these showed that the increases in performance with number of runs and over time were significantly lower in males. The final interaction term was between elevation gained and age, showing that venues with more climbing disproportionately reduced the performance of older runners.

### 3.4. The Gender Ratio of Participants at Parkrun Events in Scotland

A generalised binomial mixed model of the gender of participants identified a positive association between age and proportion of males, with an increasing proportion of males in older cohorts (Table 3, Figure 3). The model also revealed a negative association between proportion of males and parkrun ID, showing that an increasing proportion of new parkrun participants at Scottish parkrun events are female (Table 3). There was also a negative association with date, showing that the proportion of male participants is decreasing over time (Table 3, Figure 4). There was a negative association revealed between the proportion of male participants and how remote a location was, showing that females are more likely to participate at more remote event venues (Table 3, Figure 5). There was a negative association between the proportion of male participants and elevation gain, with females more likely to participate at hillier events. Men were also more likely to attend events on harder surfaces (Table 3).

### 3.5. Age of Participants at Parkrun Events in Scotland

A linear mixed model of the age of participants at Scottish parkrun events (Table 4) revealed that the average age is increasing over time, yet the age of new registrants is decreasing.

### 3.6. Performance of New Participants

A linear model of the performance of the first time each participant took part in a Scottish parkrun event produced a similar set of highly significant associations, as identified by the full model (Table 5). Number of participations was not included, as it would be 1 in all cases. A stronger gender effect was detected than in the full model, showing that males perform particularly well when they first participate compared to females (Table 5). There was a decline in the AGPS of first-time participants over time (Table 5). A similar pattern is seen with registration number if this is included instead of date in the model. Please note that variance inflation factors revealed that both should not be included together in this model, as they are too highly correlated. It is also noteworthy that this model reveals a decrease in the proportion of males as first-time participants over time.

## 4. Discussion

This study provides the first large-scale investigation of the fitness benefits of parkrun by analysing the performances of over 1.5 million participations in Scotland. Clear patterns in participation and performance at parkrun events in Scotland have been identified, which could be used to increase the beneficial impact parkrun has on communities in Scotland and beyond. Although traditional mass participation events, such as annual marathons, might not increase community activity levels [3], the weekly parkrun model does seem to provide a clear public health benefit. For example, this study reveals that performances at parkrun improve with the number of times participants attend parkrun.

There is a substantial and continuing decline in the average performance of participants identified by the study. This is not predicted if parkrun simply attracted existing runners. The decline is not the result of participants becoming less fit; indeed, this study identified that individual participants improved their performance over time, revealing a public health benefit associated with registering for parkrun. Consequently, it seems likely that the decline in overall performance is being driven by decreasing fitness levels of newer participants. This is supported by the negative association identified between registration number and performance, revealing that the most recent registrants exhibited the lowest performance. There is a clear pattern of continuing decline over time in the performance of the new registrants suggesting parkrun is becoming increasingly attractive to the less active. It seems that the parkrun model of mass participation is successfully increasing the activity levels of the less fit, and is becoming increasingly effective at doing so. The fact that participants increase their individual performance over time and with number of participations shows that parkrun is providing a public health benefit, and the impact of that benefit is likely to be increasing as parkrun becomes more inclusive.

The study also found a substantial difference of 1.76% in the performance levels of men and women. This fits with the relative patterns of inactivity associated with gender [2,4]. The lower level of starting fitness of women is also highlighted by the finding that women improve their performance faster over time and with increasing numbers of completed events than male participants. This shows that women on average benefit more than men from parkrun, something that might potentially be used to encourage more women to participate. It is also noteworthy that the gender gap was significantly wider when only the initial run of each participant was considered, showing that men perform significantly better than women on their first run, also indicating a higher initial fitness of men. It is also noteworthy that this gap is also narrowing over time, suggesting that parkrun is attracting fewer active male participants.

The gender differences in performance could be partly explained by greater competitiveness in men, but the lack of comparable gains in fitness in men suggests it is more likely that women are comparatively less fit when they join. One factor making men more competitive could be a higher engagement with the increasing gamification of parkrun. Parkrun has received criticism for ranking individual times and maintaining course records when parkruns are marketed as not being races [8]. However, these are likely to be important motivating factors for many to take part in parkrun. Furthermore, some elements of the gamification of parkrun are not biased towards those with higher fitness levels, such as parkrun tourism [25]. Additional gamification of parkrun could be targeted towards statistics that are not associated with absolute performance. Therefore, promotion of statistics related to improvements in individual performance in addition to ranking absolute performance might be more beneficial, and could increase the participation of those sections of the community that would benefit from it most. For example, these could include performance score personal bests (PBs) in addition to absolute PBs.

This study identified several factors associated with the proportion of male participants at events. Those events on faster surfaces and which were closer to other parkrun venues had more male participants. Older participants were more likely to be male, although among young adults there were more female participants. In Scotland, it appears new registrants are more likely to be female. Despite this, it is only the remotest parkrun events that have higher numbers of females than males participating. The difference in registration patterns and attendances suggest there might be greater barriers to females attending parkrun events, particularly in the older age groups. The higher level of participation of younger female adults compared to older female adults suggests that the barriers to female participation might be cultural and associated with older generations. Younger women, who are more likely to have childcare limitations, are more likely to participate, suggesting that parkrun’s strategy of allowing buggies and children has successfully removed one key barrier to female participation, namely childcare responsibilities.

Remote event venues could have a greater sense of community identity, and as a result feel more welcoming. They might also have a smaller field with fewer elite athletes, making them seem less intimidating to slower participants. The slowest mean finishing times at Scottish parkrun venues are generally at the remotest events, some of which, such as Thurso and Agnew, have developed a culture of walking [18]. The development of strong social identities from being part of the parkrun community has been found to be an important component of participation and gaining health benefits from parkrun [26,27]. If remote parkruns have a stronger social identity, then they might have a disproportionately large impact on the health of the communities they serve. The proportion of female participants was also higher at events run on trails and grass compared to hard surfaces. This could be because these parkruns are at more pleasant locations, which could preferentially attract priority groups, such as women and those with mental health issues [16]. Tarmac and concrete surfaces might attract more competitive runners and make those events feel less welcoming to the less fit. In Scotland, the tarmac and concrete events are also more likely to be in city-centre parks with larger numbers of attendees, which could also make them feel more intimidating. This might contribute to the reason why ethnic minorities, who have greater access to venues, are less likely to participate.

The strong association between the age and the gender of participants suggests that older women are the least likely to take part in parkrun. This could be a result of long-held social beliefs acting as barriers, such as the expectation that women should not participate in sports. It could be fruitful to explore factors associated with relapsing at parkrun; for example, are older women more likely to relapse after attending a single event? Another barrier to participation in physical activity is the misconception that it is potentially dangerous, especially for those with long-term health conditions. Evidence suggests that the health benefits of attending parkrun considerably outweigh any risks for most groups [11,28,29]. Indeed, the parkrun Practice Initiative is based upon the assumption that participating in parkrun is beneficial for many existing patients [10].

Male participants could also be playing an important role in driving some of the patterns identified by this study. For example, the higher proportion of female participants at events on slower surfaces could be driven by male parkrunners preferentially attending events on faster surfaces. Although male participants selecting events on the basis of their speed could generate skewed gender ratios in areas where there are several events within close proximity of each other, this would not be expected to generate the higher number of female than male participants present at remoter events, where there no other local options regarding which parkrun to attend. The average age of participants at parkrun events has started to show a consistent increase, despite the average age of those registering for parkrun declining. A key factor in explaining this apparent contradiction could be the establishment of Junior parkrun, which provides 2 km events for those aged between 4–14. Junior parkrun events were not included in the study, but participants use the same registration numbers. This is likely to have encouraged more registrations of children for parkrun. The higher number of boys participating than girls in 5 km parkruns might suggest that girls find junior parkruns less intimidating than 5 km parkruns. It is also possible that decisions by parents are impacting this trend.

One very interesting finding was the discovery that the highest female gender ratios were achieved in the 20–45 age group. This suggests that barriers to activity as a result of childcare responsibilities might have been reduced by parkrun by allowing participants to run both with buggies and with children. It is older females in the age group least likely to have childcare responsibilities who are also least likely to participate. This could potentially be a legacy of the lack of equivalent opportunities available to them to participate in physical activity when they were of child-bearing age.

One potential limitation of the study is the use of AGPS as an indicator of fitness. The ideal measure of fitness would be the maximum achievable AGPS of a participant on the day they completed an event. However, participants are not likely to run at their maximum level at each event, especially as parkrun is not marketed as a race. This study identified that between-participant variance far exceeded within-participant variance. Consequently, even though participants will not always run at their maximum level at each event, the vast majority of the time they will run reasonably close to it, and AGPS are likely to be a reasonably robust indicator of fitness. Despite the variation introduced by participants running at different levels compared to their maximum attainable performance, this study has nevertheless identified clear bidirectional patterns in performance, with individual performances improving and overall performances declining.

The study was also limited by the available data. Protected characteristics, such as ethnicity and socio-economic group, which are likely to be associated with attendance and performance at parkrun events, are not publicly available, and therefore, could not be explored in this study. There are likely to be other factors interacting with those identified in this study that are yet to be explored and could alter some of the conclusions drawn. For example, are the patterns seen in females present in all social and racial groups?

One other limitation is that this study was restricted to Scotland, and it remains to be seen whether these findings translate to other areas of the UK and beyond. Conducting similar studies in other regions would be needed to determine the generality of the findings of this study.

This study identified significant patterns in the participation of parkrun events in Scotland. Parkrun in Scotland seems to be becoming more inclusive as less fit individuals increasingly take part. The proportion of women taking part has also increased in parallel with this reduction in performance. Women are particularly willing to take part in the most remote isolated venues, so these events are likely to be having a disproportionately large beneficial impact on their local communities. Indeed, remote parkrun events have more female than male participants, suggesting that some barriers to women participating that are present at other events might be absent from remote Scottish parkrun events. Alternatively, there could be more barriers to male participation in more remote locations, or perhaps people living in remoter locations are naturally more likely to be active in their local environment and so participation levels are more even between genders. Parkrun could try to encourage the creation of events in more remote locations. It might be interesting to explore the proportion of the local population who attend these events to determine if it is higher because more women participate, or lower because fewer men do, as they have other potential options for outdoor pursuits. The creation of more events in remote locations might accentuate inequalities in access to parkruns with respect to socio-economic background; however, parkrun has identified that their locations are skewed towards providing easier access for both lower socio-economic classes and ethnic minorities [13]. This is likely due to the relatively high number of locations in inner city parks compared to more rural areas. Thus, creating more remote events might help remove the bias towards access to parkrun in urban communities. Furthermore, an increase in the number of parkruns will not reduce anyone’s existing access, but only increase access for those currently lacking a local event. Another advantage of increasing the number of parkruns in remoter areas is providing a more local event for existing parkrunners in that area, thus reducing their travelling time and the impacts of their travel on the environment.

The proportion of female participants was also substantially higher at events run on trial paths and grass. Parkrun could encourage the creation of softer surface events in areas dominated by hard surface events to provide a less competitive alternative venue for female participants. Furthermore, the parkrun practice initiative might be increasingly effective if practitioners prescribe not just attendance at a parkrun event but at an event that a new, relatively unfit, participant will more likely return to. Finally, an increase in gamification targeted towards improvements in performance could also encourage more participation by those individuals who would benefit the most.

A study of why people stop attending parkrun events would also be useful. For example, are women more likely to relapse after attending a single event? Is this more likely to happen at larger, more urban events with a greater proportion of elite athletes? Understanding what factors are creating barriers to continued participation could be as useful as studying patterns in participation itself.

## 5. Conclusions

This study revealed that parkrun is becoming increasingly inclusive, as evidenced by the continuing reduction in performance of new participants and an increasing proportion of female participants. Individual participants improve their performance, revealing the fitness benefits of participating in parkrun. Women are known to have greater barriers to engaging in sporting activity, and a recent study has revealed that this also applies to parkrun [30]. This study reveals that women are more likely to attend remoter events on slower surfaces. Some barriers still remain, with fewer women taking part in events in close proximity to other events, i.e., those in city-centre parks and on faster surfaces, suggesting that women might find these events more intimidating. Older women are also still underrepresented in the parkrun community. This study suggests the creation of events on slower surfaces might increase female participation, and reveals the importance of remote parkruns to their local communities and shows how successful they have been at encouraging female participation. These findings suggest that the parkrun practice initiative, which currently links specific medical practices to specific parkruns, might benefit from more targeted prescriptions to specific types of parkrun events to maximise the chance of continued engagement [31].

## Figures and Tables

**Figure 1 ijerph-20-03602-f001:**
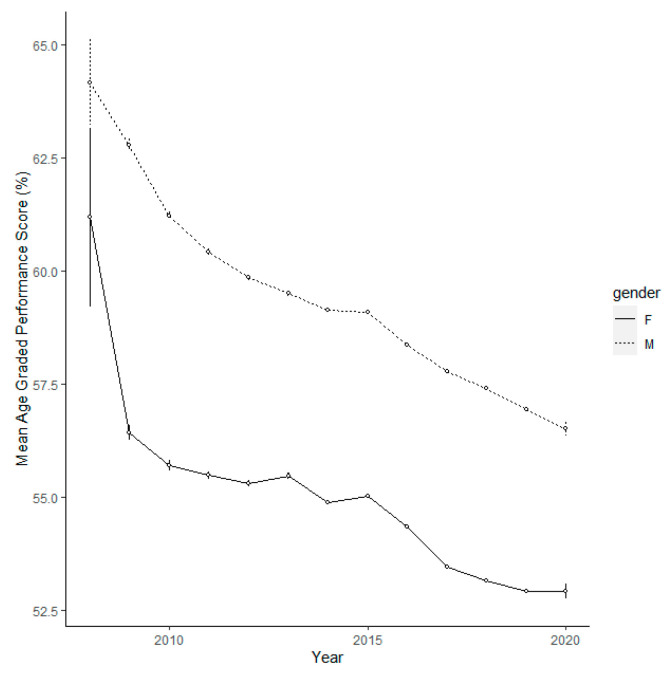
The AGPS of male and female participants separated by year at Scottish parkrun events. The error bars are 95% confidence intervals.

**Figure 2 ijerph-20-03602-f002:**
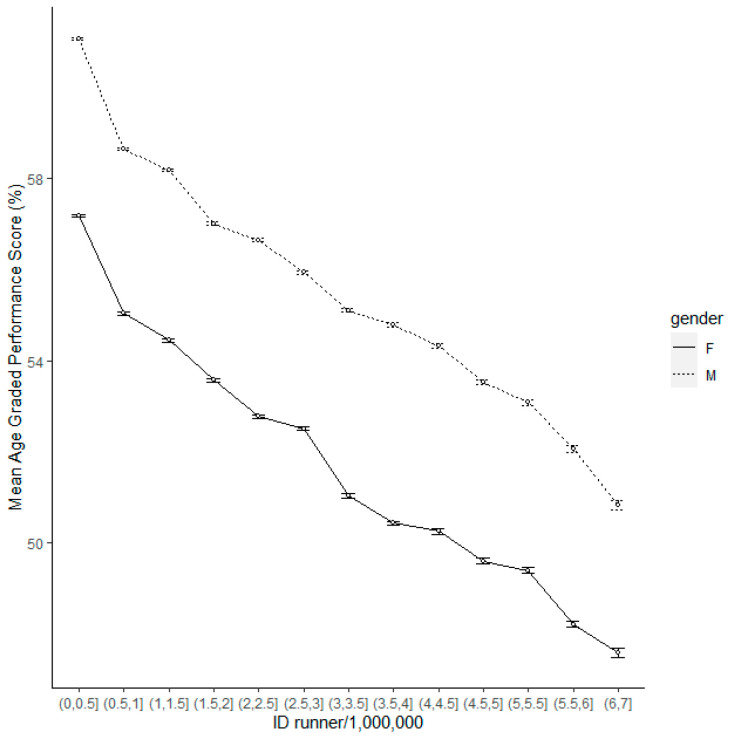
Mean AGPS for each registration ID cohort. Cohorts are separated into groups of 500,000 by ID number, i.e., ID numbers 1–500,000, 500,001–1,000,000 etc., so the further to the right of the graph, the more recent the cohort of registrants.

**Figure 3 ijerph-20-03602-f003:**
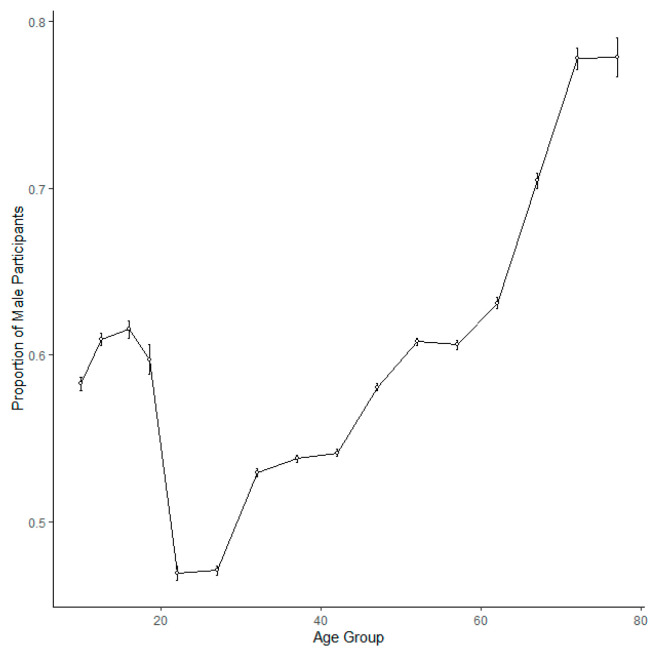
The proportion of male runners in each age cohort at Scottish parkrun events. Error bars represent 95% confidence intervals.

**Figure 4 ijerph-20-03602-f004:**
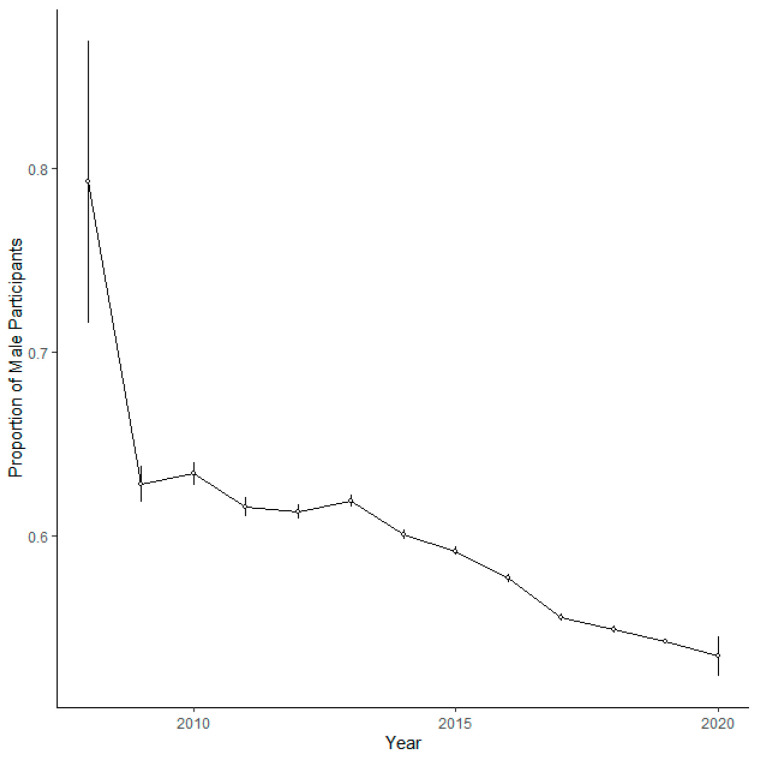
The proportion of male runners by year at Scottish parkrun events. Error bars represent 95% confidence intervals.

**Figure 5 ijerph-20-03602-f005:**
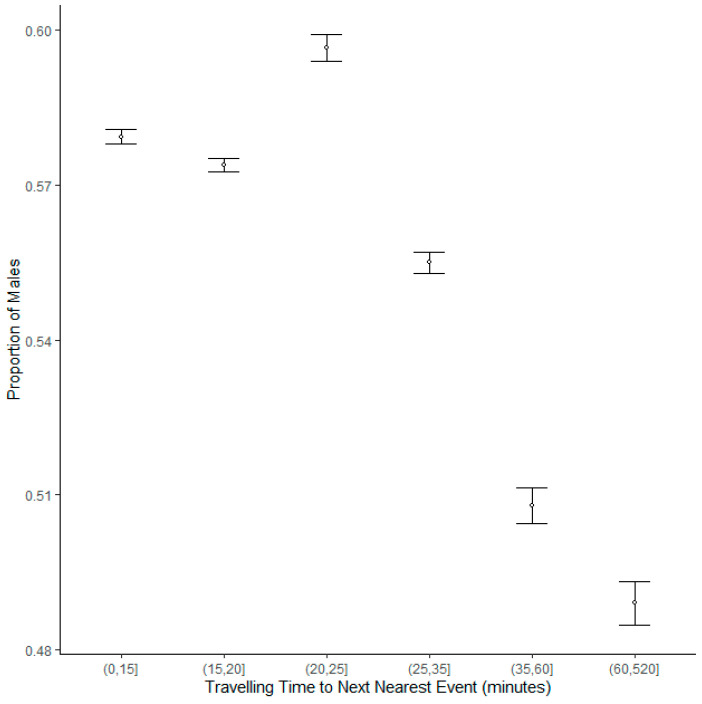
The proportion of male runners by travelling time in minutes to the next nearest parkrun event. Error bars represent 95% confidence intervals.

**Table 1 ijerph-20-03602-t001:** A list of the predictor variables included in the AGPS model.

Variable	Type	Effect	Detail
Participant	Factor	Random	A unique identifier was used for each participant
Registration number	Number	Fixed	These are allocated chronologically so a higher number corresponds to a more recent registrant
Elevation gain	Number	Fixed	The elevation gained on the route of the event
Gender (male)	Factor	Fixed	Male or female
Surface type	Number	Fixed	Hard (2), mixed hard and soft (1), soft (0)
Runs	Number	Fixed	The number of parkruns completed
Date	Number	Fixed	Date of the event
Age	Number	Fixed	Ages for participants are provided in age cohorts. The mid-value of the cohort range was used
Travelling time to next nearest venue	Number	Fixed	Travelling time in minutes to the next nearest parkrun

**Table 2 ijerph-20-03602-t002:** A linear mixed model of AGPS at Scottish parkrun events. Participant was included as a random effect. The final three parameters listed in the table are significant interaction terms that were also maintained in the model.

Parameter	*F*	*p*	Estimate	S.E.
Intercept		<0.001	55.310	0.032
Registration number	9492	<0.001	−2.029	−0.021
Elevation gain	16,388	<0.001	−0.961	0.008
Gender (male)	2894	<0.001	1.756	0.033
Surface	1814	<0.001	0.352	0.008
Runs	2358	<0.001	1.931	0.040
Date	1675	<0.001	0.282	0.007
Age	467	<0.001	0.478	0.021
Travelling time to next nearest venue	23	<0.001	−0.040	0.008
Date × gender	514	<0.001	−0.136	0.006
Runs × gender	139	<0.001	−0.446	0.038
Elevation gain × age	100	<0.001	−0.074	0.007

**Table 3 ijerph-20-03602-t003:** A generalised binomial mixed model of the gender of participants at parkrun events in Scotland. Event venue was included as a random effect. Positive parameter estimates are associated with higher numbers of male participants.

Parameter	Estimate	S.E.	*z*	*p*
Intercept	4.222	0.064	65.00	<0.001
Age	0.360	0.056	6.40	<0.001
Registration number	−5.968	0.060	99.48	<0.001
Travel time to nearest parkrun	−1.046	0.039	26.74	<0.001
Surface	0.802	0.042	18.98	<0.001
Date	−0.423	0.002	3710	<0.001
Elevation gain	−0.319	0.043	7.38	<0.001

**Table 4 ijerph-20-03602-t004:** A linear mixed model of the age of participants at parkrun events in Scotland. Event venue was included as a random effect.

Parameter	Estimate	S.E.	*t*	*p*
Intercept	36.74	0.051	714.24	<0.001
Date	2.321	0.002	1453.81	<0.001
Registration number	-2.361	0.030	79.96	<0.001
Gender (male)	-0.042	0.002	2.81	0.004

**Table 5 ijerph-20-03602-t005:** A linear mixed model of the AGPS of first timers at Scottish parkrun events. Event venue was included as a random effect.

Parameter	*F*	*p*	Estimate	S.E.
Intercept		<0.001	48.170	0.059
Gender (male)	8766	<0.001	4.289	0.048
Date	3819	<0.001	−1.531	−0.024
Surface	1972	<0.001	0.992	0.033
Age	1439	<0.001	0.894	0.024
Elevation gain	424	<0.001	−0.493	0.024
Travelling time to nearest parkrun	4.95	0.26	−0.070	0.023

## Data Availability

No new data were created, and the data belong to parkrun Global. The data were accessed as a permitted act for independent non-commercial research purposes through fair dealing legislation, allowing access to publicly available databases. Only a small proportion of the parkrun results database was accessed (data from just 56 of more than 2000 venues).

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
