# Peer review of "New Parkrunners Are Slower and the Attendance Gender Gap Narrowing Making Parkrun More Inclusive"

_ijerph, 2023, doi:10.3390/ijerph20043602_

Round 1

Reviewer 1 Report

Thank you for the invitation to review this manuscript. The paper provides a detailed analysis of data from parkrun participation in Scotland to identify factors that may influence participation. The manuscript is well-written and given the national and global presence of parkrun, parkrun represents an important health initiative that research should explore.

Abstract

1.    Line 17: Please revise the term ‘poor performance’ as this implies parkrun is associated only with performance rather than participation

Introduction

2.    Line 58: Please remove double spacing

3.    Lines 61-62: Can these data be updated to reflect the end of 2022, it is imaged these totals will have changed given parkrun’s growth

4.    Line 69: Please add a comma after ‘as women are less active’

5.    Line2 78-79: I am not sure if the phrase ‘to provide childcare whilst participating’ is the most accurate. The fact parkrun allows individuals to participate with their children, would it be more appropriate to say this means they are not required to source childcare?

6.    Lines 78-79: Can a reference be provided to support that childcare is a barrier for sport participation

7.    Lines 79-80: Can a reference be provided to support the detail regarding women having greater childcare responsibilities

8.    Line 96: Please can the author explain further what is meant by ‘gained the most from attending’ Is this in terms of improving their time, and by assumption fitness, or through other markers of physical or mental health?

9.    Line 108: Please make it clearer how ‘initial fitness’ is being defined i.e., AGPS will provide a prediction of aerobic fitness and that if this increases, this will indicate an improvement in fitness

Methods

1.  Line 127: Is ‘gender’ the correct term? Or are participants grouped based on biological ‘sex’ rather than the gender they identify as?

1.  Line 162-168: Please revise use of the term ‘gender’ as per previous comment

Results

1.  General: Please revise use of the term ‘gender’ as per previous comment

1.  Figure 1: Please add units of measurement for the y-axis. For clarity, also remove grid lines from the figure

1.  Figure 2: Please add units of measurement for the y-axis. For clarity, also remove grid lines from the figure

1.  Figure 3: For clarity, also remove grid lines from the figure

1.  Line 270: Please revise ‘N.B.’ to full word format

Discussion

1.  Line 284-298: Within this section it would be beneficial to provide some context regarding why newer registrants of parkrun are performing slower. For example, there has been increased promotion by parkrun that its not a race, encouraging walking, the recent parkwalk promotional month (October 2022), parkrun practise.

1.  Lines 301-302: it would be beneficial to explain in more detail here that improvements in fitness occur more rapidly from a lower baseline, and include supporting literature for this. This will then support the point being made regarding women improving their fitness more rapidly and starting at a lower baseline fitness

1.  Line 360: The author could note here that the parkrun practise initiative may serve to reduce this barrier

2.  Line 399: Please revise the full stop to a comma within this sentence

Author Response

Abstract

  1. Line 17: Please revise the term ‘poor performance’ as this implies parkrun is associated only with performance rather than participation

The term has been revised and the sentence restructured.

Introduction

  1. Line 58: Please remove double spacing

Thanks for pointing this out. Change made as suggested.

  1. Lines 61-62: Can these data be updated to reflect the end of 2022, it is imaged these totals will have changed given parkrun’s growth

This is the line. “There are now more than 700 parkrun event venues in the UK and over 2000 worldwide with Scotland hosting 56 weekly parkrun events by the end of 2019.”

Both the 700 and 2000 are accurate and appropriate. There are now 67 events in Scotland, however the study only considered events at the 56 that were present by the end of 2019 so this statement is accurate for the study period and providing a different number might be misleading.

  1. Line 69: Please add a comma after ‘as women are less active’

Thanks for pointing this out. Change made as suggested.

  1. Line2 78-79: I am not sure if the phrase ‘to provide childcare whilst participating’ is the most accurate. The fact parkrun allows individuals to participate with their children, would it be more appropriate to say this means they are not required to source childcare?

This sentence has been rewritten.

  1. Lines 78-79: Can a reference be provided to support that childcare is a barrier for sport participation

Yes, I should have added a reference here. Now done. Thanks.

  1. Lines 79-80: Can a reference be provided to support the detail regarding women having greater childcare responsibilities

Done

  1. Line 96: Please can the author explain further what is meant by ‘gained the most from attending’ Is this in terms of improving their time, and by assumption fitness, or through other markers of physical or mental health?

Yes, that did need more clarification. Now edited.

  1. Line 108: Please make it clearer how ‘initial fitness’ is being defined i.e., AGPS will provide a prediction of aerobic fitness and that if this increases, this will indicate an improvement in fitness

Thanks for that. I used the term fitness when performance would have been more appropriate which resulted in the query. So I have removed the term fitness and consistently used the term performance.

Methods

  1. Line 127: Is ‘gender’ the correct term? Or are participants grouped based on biological ‘sex’ rather than the gender they identify as?

Yes, parkrun use the term gender. New registrants select their gender. A small number of participants choose not to select a gender. These were excluded on the basis of their gender choice. So gender is the correct term.

  1. Line 162-168: Please revise use of the term ‘gender’ as per previous comment

Answered above.

Results

  1. General: Please revise use of the term ‘gender’ as per previous comment

Answered above

  1. Figure 1: Please add units of measurement for the y-axis. For clarity, also remove grid lines from the figure

Thanks for pointing out the missing units. Now added and gridlines removed.

  1. Figure 2: Please add units of measurement for the y-axis. For clarity, also remove grid lines from the figure

Thanks for pointing out the missing units. Now added and gridlines removed.

  1. Figure 3: For clarity, also remove grid lines from the figure

Gridlines removed. Gridlines also removed from Fig 4 and Fig 5.

  1. Line 270: Please revise ‘N.B.’ to full word format

Done.

Discussion

  1. Line 284-298: Within this section it would be beneficial to provide some context regarding why newer registrants of parkrun are performing slower. For example, there has been increased promotion by parkrun that its not a race, encouraging walking, the recent parkwalk promotional month (October 2022), parkrun practise.

I think it would be really interesting to see how each of these initiatives have impacted the demographics of parkrun. Indeed this is something that could be tested within a model. However, the current dataset is not the best placed to address these hypotheses. The parkwalk promotion month occurred nearly three years after the end of the study period so it would not have impacted the results. I don’t have any record of precisely when parkrun started requesting run directors state that parkrun is not a race however I’ve found discussion of this activity on social media going back to at least 2012 which is before the vast majority of the events studied here took place. I suspect that the “not a race” tag appeared at the of the rebranding from Time Trials.to parkruns. At this point only a   single event existed in Scotland. The parkrun practice initiative wasn’t started until 2018 which is towards the end of the study period. I know only some parkruns are partnered with a practice but I don’t know which. So I don’t think there’s much evidence that any one these initiatives has directly impacted the findings as things stands but I think it would be well worthwhile exploring this further in other datasets more appropriate for addressing these intriguing questions. My best guess is that word of mouth about parkun being welcoming to walkers and slower runners has been the main reason for the continued decline seen in the current study, but it is just that, a guess, so I would prefer to leave speculation on this subject for and potentially address these questions more robustly in the future.

  1. Lines 301-302: it would be beneficial to explain in more detail here that improvements in fitness occur more rapidly from a lower baseline,and include supporting literature for this. This will then support the point being made regarding women improving their fitness more rapidly and starting at a lower baseline fitness.

I’ve added the parameter estimate for the difference in the genders of 1.76%. The interaction term between gender and day measures the difference in the rate of gain of performance by the two sexes. Readers can refer back to the model. I think it is better not to repeat too many statistics in a Discussion. There’s no citation to be included here as this is one of the key novel findings of the study.

  1. Line 360: The author could note here that the parkrun practise initiative may serve to reduce this barrier

An excellent observation! A new sentence added as suggested.

  1. Line 399: Please revise the full stop to a comma within this sentence

Thanks for pointing this out. Change made as suggested.

Reviewer 2 Report

Interesting paper.

Need to add a theoretical section and provide a better literature review.

There are too many repetitions.       

Author Response

Need to add a theoretical section and provide a better literature review.

The reviewer statement about adding a theoretical section is brief and vague. Published papers on parkrun rarely contain a theoretical section and then when they do it is because the study is testing a specific component of existing theory. This study is not testing the predictions of any existing theory but developing novel insights from the parkrun results database.

The literature review is extensive and covers a considerable proportion of the published papers on parkrun. No specific details have been provided about what sections of the literature.

There are too many repetitions. 

I’m not aware of any repetition in the paper nor does the reviewer provides specific details about what is being repeated or where.

Reviewer 3 Report

General Comments

The aim of this study is to identify characteristics of events that overcome barriers to participation and identify changing patterns in the demographics of participants. The research submitted is well designed, of great importance and innovative in its execution. However, when it comes to explaining what has been done, I have found certain shortcomings that should be improved to be published. These considerations are included in the specific comments.

I kindly ask the authors to read this report carefully and to respond accurately to the suggestions made if they consider them appropriate. I would like to thank you for your time and investment in this article and consider this review in the best possible way to improve and make science better.

Specific Comments

Material and Methods

The method section should be subdivided for better understanding of the numerous variables and data extraction performed. At the same time, I consider the inclusion of tables or figures that help to better understand the criteria used to be very positive.

The inclusion criteria should be better defined.

Author Response

The method section should be subdivided for better understanding of the numerous variables and data extraction performed. At the same time, I consider the inclusion of tables or figures that help to better understand the criteria used to be very positive.

The inclusion criteria should be better defined.

This is an accurate observation by the reviewer. I have substantially revised the materials and methods in line with the reviewer’s suggestions. The section has now been broken down into subsections, the descriptors for the predictor variables revised and I have included the reviewer’s excellent suggestion of a table of predictor variables. That is a delightfully simple solution that adds considerable clarity to what was included in the model.